# Recent Antimicrobial Responses of Halophilic Microbes in Clinical Pathogens

**DOI:** 10.3390/microorganisms10020417

**Published:** 2022-02-11

**Authors:** Henciya Santhaseelan, Vengateshwaran Thasu Dinakaran, Hans-Uwe Dahms, Johnthini Munir Ahamed, Santhosh Gokul Murugaiah, Muthukumar Krishnan, Jiang-Shiou Hwang, Arthur James Rathinam

**Affiliations:** 1Department of Marine Science, Bharathidasan University, Tiruchirappalli 620024, India; henzmarine@gmail.com (H.S.); tdvengatesh7@gmail.com (V.T.D.); msgokul89@gmail.com (S.G.M.); 2Department of Biomedical Science and Environmental Biology, Kaohsiung Medical University, Kaohsiung 807, Taiwan; hansd@kmu.edu.tw (H.-U.D.); johnthinimuneerahmd@gmail.com (J.M.A.); 3Department of Physics, National Institute of Technology, Tiruchirappalli 620015, India; marinekmk@gmail.com; 4Institute of Marine Biology, National Taiwan Ocean University, Keelung 20224, Taiwan; 5Center of Excellence for Ocean Engineering, National Taiwan Ocean University, Keelung 20224, Taiwan; 6Center of Excellence for the Oceans, National Taiwan Ocean University, Keelung 20224, Taiwan

**Keywords:** antibiotic resistance, salinity, halophilic, bioactive compound, pharmaceutical

## Abstract

Microbial pathogens that cause severe infections and are resistant to drugs are simultaneously becoming more active. This urgently calls for novel effective antibiotics. Organisms from extreme environments are known to synthesize novel bioprospecting molecules for biomedical applications due to their peculiar characteristics of growth and physiological conditions. Antimicrobial developments from hypersaline environments, such as lagoons, estuaries, and salterns, accommodate several halophilic microbes. Salinity is a distinctive environmental factor that continuously promotes the metabolic adaptation and flexibility of halophilic microbes for their survival at minimum nutritional requirements. A genetic adaptation to extreme solar radiation, ionic strength, and desiccation makes them promising candidates for drug discovery. More microbiota identified via sequencing and ‘omics’ approaches signify the hypersaline environments where compounds are produced. Microbial genera such as *Bacillus*, *Actinobacteria*, *Halorubrum* and *Aspergillus* are producing a substantial number of antimicrobial compounds. Several strategies were applied for producing novel antimicrobials from halophiles including a consortia approach. Promising results indicate that halophilic microbes can be utilised as prolific sources of bioactive metabolites with pharmaceutical potentialto expand natural product research towards diverse phylogenetic microbial groups which inhabit salterns. The present study reviews interesting antimicrobial compounds retrieved from microbial sources of various saltern environments, with a discussion of their potency in providing novel drugs against clinically drug-resistant microbes.

## 1. Introduction

Clinical sectors have been confronted with health risk challenges provided by antibiotic resistance (ABR). This phenomenon warrants the development of effective antibiotics, particularly against human pathogens that cause serious threads [1]. The usage of large antibiotics for human therapy as well as for animals, such as those important for agriculture andaquaculture, results in a selection of pathogenic microbes that are resistant to multiple drugs. Regional surveillance of ABR, including Africa, America, Europe, the eastern Mediterranean and the southeastern and western Pacific, was highlighted by the WHO with respect to specific pathogens, such as *E. coli* (third-generation Cephelosporine resistance), *K. pneumoniae* (Carbapenems and third-generation Cephelosporineresistance), *S. aureus* (methicilllin resistance), *Shigella* sp. (resistant to fluoroquinolones) and *Neisseria gonorrhoeae* (susceptibility decreasing to 3rd generation Cephelosporines) [2]. Every year, at least 2.8 million people in the United States become sick with antibiotic-resistant bacteria or fungus, and over 35,000 people die as a result [3]. Recently, due to the emergence of ABR, progress reports were prepared by the CDC for the years 2016–2020, and various health practices, including usage of drugs for human and veterinary health, were implemented and signed by the respective partner countries [4]. ABR is also known to cause infections associated with healthcare facilities and is likely to betransferred between healthcare facilities. Emerging technologies are currently being used to eradicate drug-resistant strains by utilising diverse medicines and functionalised biomaterials. However, not only in the human sector but also in the animal and environmental sectors is the phenomenon of drug resistance difficult to overcome [5]. As for the cell walls of bacteria, selective mechanisms emerged, such as membrane permeability, efflux pumps, and the alteration of target molecules modifyingcell wall precursors, resulting in drug resistance. Biomolecule discovery necessitates the development of novel drugs to address these difficulties [6]. Microorganisms from extreme habitats have recently attracted a lot of attention. This is mostly related to the evolution of molecular components in their living systems, as well as the stability of macromolecules [7]. Halophilic microorganisms are salt-tolerant extremophiles that thrive at high salt concentrations. Recent research indicates that halophilic or halotolerant bacteria and fungi from high saline environments provide a suitable source of biosurfactants, enzymes, and aromatic chemical degraders [8,9,10]. Hypersaline regions offer several possibilities for the synthesis of secondary metabolites withbioactivities of industrial interest [11]. Several haline lakes, salterns accommodating microbes such as *Pisibacillus* and *Nocardiopsis* possess antibacterial activity by excreting its potential extract and compounds, such as pyrrolo (1,1-A(pyrazine-1,4-dione,hexahydro-3-(2-methylpropyl)-) [12,13]. Cold environments, such as the Antarctic Casey station, also support halophilic bacteria that create lipopeptides with enzymatic and antibiotic properties of applied interest [14]. Arctic subsea sediments of the Barents Sea harbour *A. protuberus*, which produces the antifungal compound Bisvertinole [15]. The halophilic bacterium *Vibrio azureus* MML1960 was found to have antifungal action against fluconazole-resistant *Candida albicans* [16]. A metabolite secreted by the halophilic *Pseudomonas aeruginosa* developed an antibiotic against methicillin-resistant *S. aureus*. [17]. Certain archaea, such as *Haloquadratum walsbyi* and the bacterium *Salinibacter*, have different anaerobic growth, produce gas vesicles, and deliver halocins able to kill other archaea, and certain strains produce pigments such as carotenoids with various strong bioactivities, including antioxidants [18,19,20,21]. Some enzymes and proteins generated by halophiles were tested for antibacterial activities against plant diseases, including L-asparaginase, amylase, protease, lipase, cellulose, and glycoproteins from *Halomonas* and *Bacillus* spp. [22,23]. The bioactivity of halophilic microbes from diverse saline environments against various pathogens has increased interest in biomolecule applications in the pharmaceutical industry. Moreover, many prospective bioactivities of halobacteria, halofungi, haloarchaea, and halo-diatoms remain unexplored [24]. More attention should be paid to halo-microbial communities as a reliable source of novel drugs against drug-resistant bacteria. Consequently, the current review emphasises the recent antimicrobials produced by halophilic microbes against clinical drug-resistant strains and discusses the adaptation strategies of halophiles for extreme environments.

## 2. Halophiles: A Potential Source of Antimicrobials

There are more examples of hypersaline locations throughout the world, such as coastal lagoons, soda and salt lakes, hypersaline human-made ponds for salt production (salterns), deep sea brine pools (formed by salt dissolution during seafloor tectonic activity), brine channels in sea ice, and brine pickling solutions. Both halophiles and halotolerants produce antimicrobials at optimal culture conditions, such as the halophilic *Actinomycetes* sp., halophilic *Kocuria* sp., and halotolerant *Micromonospora* sp., which secrete antibacterial compounds against *Staphylococcus citreus*, *Staphylococcus aureus*, and *Vibrio cholera* [25]. Even some antifungal activities were provided by hypersaline actinomycete genera against *Aspergillus niger*, *Cryptococcus* sp., and *Fusarium solani* [26]. Antimicrobials derived from *Microbacterium oxydans* and *Streptomyces fradiae* of foreshore soils showed broad-spectrum action against *P. aeruginosa*, *S. typhi*, *Micrococcus luteus, C. albicans*, and *Colletotrichum gloeosporioide* [27]. Ultimately, the principal phylum responsible for the inhibition of clinical pathogens are Actinobacteria, which are available as frequent isolates from solar salterns, sea floor sediments, and mangroves. The predominant genera here are *Streptomyces* and *Nocardiopsis* [26,28,29]. Aside from the Actinomycetes, other genera, such as *Bacillus* (*Bacillus* sp. BS3) and *Vibrio* (*Vibrio parahaemolyticus*), have previously been identified as antimicrobial producers against human pathogens, such as *E. coli*, *P. aeruginosa*, *S. aureus*, and *B. subtilis*, as well as *S. albus* [30,31] (Table 1). Furthermore, an ethyl acetate extract of the solar saltern *Halomonas salifodinae* bacteria exhibits antibacterial action against aquatic pathogens, such as *Vibrio parahaemolyticus*, *Vibrio harveyi*, *Aeromonas hydrophila*, and *Pseudomonas aeruginosa*, isolated from fish and shrimp [32]. The purified fraction of the aforementioned metabolites has antiviral efficacy and contains compounds, such as Perfluorotributylamine, Cyclopentane, 1-butyl-2-ethyl and 1,1′-Biphenyl]-3-amine, Pyridine, 4-(phenylmethyl)-Hexadecane, 2-methyl-, and Nonandecane, which suppresses the replication of white spot syndrome virus (WSSV) in *Fenneropenaeus indicus*. The domain Archaea contains 56 genera and 216 species of procaryotes that produce halocin (antimicrobial peptides) [33]. The archaea *Haloferaxlarsenii* HA3 has cross-domain antibacterial action and inhibits the growth of *H. larsenii* HA10 [34,35]. Furthermore, the supernatant of the halocin-synthesising strain *Haloferaxmediterranei* DF50-EPS (incapable of making EPS (Exopolysaccarides)) induces DNA uptake, as evidenced by the uptake of the pWL502 plasmid [36]. The Halocin C8 peptide (7.4 kDa) generated by *Natrinema* sp. AS7092 strongly inhibits *Halorubrum chaoviator* [37]. However, there is no potential evidence that halocins are effective against human pathogenic microorganisms. Chemical molecules from halophilic microorganisms, such as indole derivatives, alkaloids, tripenoids, and peptides, showed some bioactivity against certain pathogens [38]. Several bacterial genera isolated from halophilic ecological environments produce antibiotic compounds that are effective against various pathogens. Figure 1 depicts the phylogenetic representation of antimicrobial agents producing halophilic bacterial strains generated from recent literature using MEGA –X Software [39]. Other than bacteria, a halophilic fungus *Aspergillus protuberus MUT 3638*, isolated from Arctic Ocean abyssal marine sediments, has antibacterial effectiveness against *A. baumanii, B. metallica, S. aureus*, and *K. pneumoniae* [15]. Antibacterial and antioxidant capabilities are found in *Aspergillus gracilis, Aspergillus penicillioides*, and *Aspergillus flavus* [40]. The marine diatoms *Chaetoceros pseudocurvisetus* and *Skeletonemacostatum* have been studied lately for their anti-tuberculosis action, particularly under phosphate-depleted circumstances, with non-toxic effects on human cell lines [41]. The Amberlite resin extract of *Chaetoceros pseudocurvisetus* at 800 g/mL inhibited the growth of *Mycobacterium tuberculosis* by 99%. *Skeletonema costatum* also demonstrated antifungal and antibacterial activity against *Fusarium moniliforme* and *Streptococcus pyogenes* with 18 mm diameter of inhibition zones via methanol and ethanol extracts, and other diatoms, including *Chroococcusturgidus*, revealed a significant inhibition zone against *E. coli* with 21.4 mm diameter via methanol and ethanol extracts [42].

## 3. Biopotency of Halophiles as Antibacterials for Clinical Drug-Resistant Pathogens

Drug resistance in clinical strains is updated against antibiotics in both Gram-positive and Gram-negative strains, such as Enterobacteriaceae (Cephalosporines- and Carbapenem-resistant), *Pseudomonas aeruginosa*, and *Neisseria gonorrhoeae* (Aminoglycosides- and quinolone-resistant) *Helicobacter pylori* (Clarithromycin), *Haemophilus influenza* (ampicillin), and *Staphylococcus aureus*, a highly infectious strain to humans with resistance to methicillin (MRSA) and intermediate to vancomycin, Enterococcus faecium (vancomycin- and cephalosporin-resistant), and *Streptococcus pneumoniae*(penicillin-resistant) [58]. Surprisingly, the quorum sensing (QS) of *P. aeruginosa* caused fluconazole resistance in *Candida albicans* by generating QS component N-(3-Oxododecanoyl)-L-homoserine lactone via the reverse pathway of ergosterol production [59]. To address these concerns, the use of halophilic biomolecules against drug-resistant bacteria has recently gained attention, particularly since novel anti-MRSA drugs were discovered (Figure 2). The halophilic bacterium *Vibrio azureus* MML1960 from saltpan sediments (Kelambakkam saltpan) attributed anti-candidal activity on fluconazole-resistant *Candida albicans*, with 0.375 mg/mL of its crude extract with a maximum inhibition zone of a 26 mm diameter [16]. Furthermore, *Vibrio* sp. A1SM3-36-8 was found in Colombian solar salterns to be the producer of 13-cis-docosenamide, a unique antibacterial agent against MRSA [53]. The halophilic *Bacillus* provides a significant amount of bioactive molecules. However, the majority of them were thought to be anticancer agents rather than antimicrobials [38]. *Bacillus firmis* VE2, a halophilic bacterium isolated from Vedaranyam sediments, produced Subtilisin ‘A’, a protein with antifungal activity against *C. albicans* and *C. parapsilosis* with 15 mm diameter of inhibition zones, as well as *S. aureus* with 16 mm [60]. The Batim and Ribandar saltpans with *Bacillus* and *Virgibacillus* spp. produced metabolites against both MRSA and MSSA (methicillin-sensitive *S. aureus*) with more than 20 and 18 mm diameter inhibition zones [61]. Halophilic *P. aeruginosa* shows antibacterial activity against MRSA with MIC (Minimum Inhibition Concentration) at 250 µg/mL [62,63]. The ethyl acetate extract of halophilic *P. aeruginosa* isolated from coastal saltpan sediments exhibits broad antibacterial activity against Norfloxacin and Ciprofloxacin-resistant *Klebsiella quasivariicola*, vancomycin-intermediate *E. coli*, and methicillin, as well as Norfloxacin-resistant *S. argenteus* isolates from diabetic foot infections, with inhibition zones with diameters of 24, 21, and 22 mm [64]. The halophilic actinomycete, *Nocardiopsis* sp. HR-4, recovered from the soil of a Salt Lake in the Algerian Sahara, offers greater antimicrobials against drug-resistant bacteria. It produces a novel natural product,7-deoxy-8-O-methyltetrangomycin, which is effective against MRSA (ATCC 43300) [54]. *Nocardiopsis* sp. JAJ16 isolated from Crystallizer Pond and *Nonomuraea* sp. JAJ18 from Indian coastal solar salterns also provide antibacterial activity against MRSA [65,66]. *Marinispora* sp. NPS12745, isolated from marine sediments in Mission Bay, southern California, produces Lynamicin E, which has antibacterial action against penicillin-resistant *Streptococcus pneumoniae* ATCC 51915, vancomycin-sensitive *E. faecalis* ATCC 29212, and vancomycin-resistant *E. faecium* [67], and *Streptomyces* sp. CNQ-418 from marine sediments of La Jolla, California, produces the compounds Marinopyrroles A and Marinopyrroles B, which were also active against MRSA [68]. Substantially, the endophytic *Streptomyces* SUK-25-derived compound DKPs cyclo-(l-Val-l-Pro), cyclo-(l-Leu-l-Pro) and cyclo-(l-Phe-l-Pro) provoked bioactivity against MRSA and *Enterococcus raffinosus* [69]. Isolates from the coasts of Papua New Guinea Bismarck and the Solomon Sea, such as *Micromonospora nigra* DSM 43818, *Micromonospora rhodorangea*, and *Micromonospora halophytica* DSM 43171, demonstrated bioactivity against several Gram-positive MDR strains, vancomycin-resistant enterococci, and MRSA [70]. *C. albicans* was also inhibited by halophilic actinobacterial strain H262 from Algerian arid habitats of the Sahara desert with a 17 mm inhibition zone, 19 mm for *Penicillium expansum* fungi PE1, 31 mm for the bacterium *B. subtilis*, and 37 mm for MRSA [71]. Moreover, Gohel et al. (2015) [72] provided a thorough description of the antibacterial activity of haloalkaliphilic actinobacteria. Extensively, halophilic Proteobacteria have already been shown to synthesise a variety of natural compounds [73]. The marine alpha Proteobacteria *Labrenzia* spp. synthesised cyclopropane fatty acids with broad antimicrobial action against MRSA and the fungus *Eurotium rubrum* DSM 62631 [74]. A study conducted in Yuncheng Salt Lake, China, investigated potential halophilic strains, such as 3, 6, 15, 12, 15, and 16, belonging to different families, such as the Clostridiaceae, Staphylococcaceae, and Bacillaceae, that inhibit the growth of *S. aureus*, *E*. *coli*, *C. albicans*, *F. moniliforme*, *F. semitectum*, and *F. xysporum* [23]. These reports also state that by using halo microbial compounds, most drug-resistant strains are rendered less virulent.

## 4. Recent Activity Findings from Halophiles—Against Clinically Important Pathogens

### 4.1. Halophilic Bacillus sp.

*Bacillus* and *Virgibacillus* were frequently isolated from saline systems with antimicrobial potential [75]. *Bacillus pumilus* NKCM 8905 *Bacillus pumilus* AB211228 isolates of coastal soil, Arabian Sea, Mumbai, produced antibiotics against *E. coli*, *S. aureus*, *B. subtilis* and *A. niger* [76]. Phospholipid compounds produced from halophilic *B. subtilis* had a better antimicrobial activity than alkaliphilic *B. subtilis* on *S. aureus* with a maximum of 26 mm diameter inhibition zone, whereas alkalic *Bacillus* sp. showed 21 mm [77]. *B. subtilis* derived from Haj Aligholi Salt Deserts and Dagh Biarjmand, Iran, revealed antimicrobial activity against pathogenic fungi and bacteria with MIC ranges from 12.5 to 25 μg/mL, fungus includes *A. flavus*, *F. oxysporum*, *C. albicans*, and the bacterium includes *B. cinerea*, and *N. crassa* with inhibition zones with diameters of 14, 11, 8, 39, and 13 mm [75]. *B. subtilis* isolated from Kovalam Beach waters, Chennai in India, shows activity against clinical pathogens *P. aeruginosa*, *Proteus mirabilis*, *K. pneumonia*, *Salmonella typhi* and *S. typhi* B. The chloroform crude extract of this bacterium containing compound Pyrrolo (1, 2-a) pyrazine-1, 4-dione might be responsible for the reduction in OD (optical density) compared to the control for the aforementioned bacterial species [78]. *Bacillus persicus* 24-DSMisolated from Dead Sea mud provided activity against *B. subtilis* and *E*. *coli* [79]. Another discovery revealed that the *Bacillus* species DSM2 from the same location has activity against pathogenic fungi, including *C*. *albicans* ATCC 10231 and *A. brasiliensis* ATCC 16404 (Maher 2017) [80].

### 4.2. Halophilic Actinomycetes

Due to the wide range of biopharmaceutical applications of Actinobacteria, there is a great diversity of halophilic strains being studied [81]. *Nocardiopsis dassonvillei* halophilic actinomycetes showed antibacterial efficacy against human pathogens, such as *S. aureus*, *E. coli*, *B*. *cereus*, and *P. aeruginosa* [82]. The ethyl acetate extracts of *Kocuria* sp. strain rsk4 inhibit *S. aureus* at the lowest MIC of 30 g/mL by secreting an antibacterial unknown compound with a molecular mass of 473 g/mol [83]. The phenolic extracts of the halophilic actinomycetes isolate GD3007 provided activity at 50 µL/g against different pathogens such as *E. coli*, *S. aureus*, *Vibrio* sp., *P. aeruginosa*, and *K. pneumonia* with inhibition zone diameters of 30, 27, 24, 25, and 26 mm [84]. Corum salterns actinomycetes were found to be active against *B*. *subtilis*, *E. coli*, and *A. niger*. The most significant activity was obtained from strains belonging to *Streptomyces* providing gene clusters including PKS-I, PKS-II, and NRPS, which were also tested for antibacterial efficacy using similar primers [85]. *Streptomyces* sp. MA05, which was isolated from a salt lake in Chennai, showed antibacterial activity against *S. aureus* with an inhibition zone larger than 15 mm [86]. *Streptomyces* spp. AJ8 was isolated from the Kovalam solar saltern in India, with a single gene fragment of NRPS length and was found to have antagonistic properties against bacterial and fungal pathogens, such as *V. harveyi* (9.2 mm inhibition zone), *A. niger* (9.8 mm), and *C. albicans* (5 mm) [87].

### 4.3. Other Halophilic Bacterial Species

Other Halomonas taxa isolated from the salty habitat of Northeastern Algeria showed broad antifungal activity against *Fusarium oxyporum*, *Botrytis cinerea*, *Phytophthora capsici*, and *F. verticillioides* [88]. Gamma Proteobacteria from coastal solar salterns, such as *Halomonas smyrnensis* and *Halomonas variabilis*, were found to have antibacterial properties against *S. pasteuri* and *E. coli*. *Salinicoccus roseus* and *Virgibacillus salaries* exhibited activity against *M. luteus*, *A. johnsonii*, *X. oryzae*, *C. lipolytica*, *S. cerevisiae*, and *M*. *luteus*, *X. oryzae*, *C. lipolytica*, *S. cerevisiae* [89]. The cell supernatants of *Nocardioides* sp. of halo-Antarctic soils containing glycolipids and/or lipopeptides provided antimicrobial activity against *S. aureus* and *X. oryzae*, whereas its salt medium supplemented with various carbon sources provided enzymatic activity [14].

### 4.4. Halophilic Microalgae

*Dunaliella salina* alone produced several compounds with antimicrobial potencies against several pathogens. Hexane extract of the microalga *Dunaliella salina* at 97.0 mg mL^−1^ concentration showed an inhibition zone with a diameter of 20 mm against *B. subtilis*, and ethanolic extract at 214.0 mg mL^−1^ showed 21 mm against *B. subtilis* [90]. The methanol and chloroform extract of *Dunaliella salina* possesses antibacterial activity on *Vibrio cholerae* at a maximum 10.4 mm inhibition zone due to the unique compounds such asn-Hexadecane (M.W. 226.2) and 3, 3, 5-Trimethylheptane (M.W. 142.2) [91]. A mixed culture technique using marine and freshwater microalgae, such as *Coelastrum* sp., *Scenedesmus quadricauda*, and *Selenastrum* sp., exhibited growth inhibition on *S. epidermidis*, *S. marcescens*, and *P. fluorescens* via their methanol and hexane extracts [92]. Jafari et al., 2018 [93] proved the antibacterial efficacy of *D. salina* by suppressing the growth of *S. mutants* at 6250 g mL^−1^ using methanol, chloroform, and acetone extracts.

## 5. Novel Antimicrobials and Their Producing Strains from Halophiles

Interestingly, the novel bacterium *Paenibacillus sambharensis* isolated from a salt lake suppressed the growth of *S. aureus* by producing the compound bacitracin A, with a molecular mass of 1421.749 Da [94]. WT6 and R4A19 antimicrobials generating strains were recently retrieved from an Iranian Salt Lake, producing activities against *E. coli* and *B. cereus* [95]. The novel halophilic isolates AH35 and AH10 of the Algerian Sahara showed antibacterial activity (13–45 mm) against *K. pneumoniae*, *Pseudomonas syringae*, and *Agrobacterium tumefaciens*, and AH35 was active against *Salmonella enterica* (13 mm). The phylogenetic clades of these potential strains represent the species *Saccharomonospora paurometabolica*, *Saccharomonospora halophila*, and *Actinopolysporairaqiensis* [96].

The unexplored deep-sea habitats of the Andaman and Nicobar Islands provided a source of novel halophilic species, including Bacilli, Alpha-, and Gamma-Proteobacteria, with antibacterial activity against Gram-positive and Gram-negative strains, including *P. mirabilis* MTCC1429, *V. cholerae* MTCC3904, *K. pneumonia* MTCC109, *E. coli* MTCC443, and *S. pneumoniae* MTCC1935 [97]. The partially purified biosurfactants produced from halophilic strains (Khewra Salt Mines, Pakistan) *Halobacilluskarajiensis* and *Alkalibacillusalmallahensis* suppressed the growth of *K. pneumoniae* (94%) and *A. flavus* (80.4%) [9]. A novel p-terphenyl 1 and a novel p-terphenyl derivative 3 providing a benzothiazole moiety were discovered from halophilic *Nocardiopsisgilva* YIM 90087, thus p-terphenyl 1 signifies its activity against *F. avenaceum*, *F. graminearum*, and *F. culmorum* with 8, 6, and 128 µg/mL MICs. Compound 1 exhibits antifungal activity with MIC 32 μg/mL against *C. albicans*, *B. subtilis* with 64 μg/mL, Novobiocin 4 showed antibacterial efficacy against *B. subtilis* with 16 μg/mL MICs and *S. aureus* with 64 μg/mL MICs [98]. Despite the fact that the saline environment produces antimicrobials, some saline niches still remain unexplored and warrant urgent study for the discovery of novel antimicrobials and other bioactivities of applied interest.

## 6. Halo-Microbial Derived Products as Antimicrobials

### 6.1. *Pigments*

A type of carotenoids, bacterioruberin, was retrieved from the halophilic bacterial species *Salinicoccussesuvii* MB597, *Aquisalibacillus elongatus* MB592, and *Halomonasaquamarina* MB598, which were isolated from the salt range of Khewra, Pakistan, and provided antimicrobial activity against some pathogenic bacteria. Here, *Enterococcus faecium* was suppressed by a maximum inhibition zone diameter of 23 mm, besides wide antifungal activity attained from a pigment derived from *Halomonasaquamarina* MB598 with 98% growth inhibition on *Aspergillus fumigatus* and pigments derived from *Aquisalibacillus elongatus* MB592 showing 96% growth inhibition against the same fungus. Pigment derived from *Salinicoccussesuvii* MB597 gave 96.7% growth inhibition against *Mucor* spp. [9]. Red pigment produced by the bacterium *Candidatus chryseobacterium massiliae* isolated from Arabian seawater samples showed higher antibacterial activity among the isolated strains against *B. cerus* (8 mm), *S. aureus* (6 mm), *B. megaterium* (7 mm), *B. subtilis* (6 mm), and *V. cholerae* (8 mm) [99]. A crude extract of bright yellow pigment produced from marine *Brevibacterium* showed antibacterial activity against *S. aureus* (29 mm), *E. coli* (17 mm), *P. aeruginosa* (27 mm), and *B. subtilis* (28 mm) [100]. *Salinococus* sp. isolated from the Nellore sea coast produced a pinkish orange pigment, and its crude extract revealed antimicrobial activity against *K. pneumoniae, P. aeruginosa*, and *S. aureus* with the respective inhibition zone diameters: 16 mm, 14 mm, and 24 mm [101]. In addition, an interesting study says the prodigiosin pigment extracted from marine *Serratia rubidaea* RAM Alex strain with textile fabric coating showed antibacterial activity against *S*. *aureus* and *E. coli*, which significantly decreased the hospital-acquired infections (HAI) [102]. Marine *P. aeruginosa* producing pyocyanin was shown to act as an anti-chlamydial agent at a concentration of 0.02 µM [103]. Nanomelanin derived from *P. aeruginosa* obtained from the marine sponge *T. citrine* had antibacterial activity against *B. subtilis*, *S. aureus*, and *E.coli* [104]. Marine-derived *V. ruber* DSM 14379 producing prodigiosin showed strong killing efficiency on *B. subtilis* [105]. Marine *Streptomyces* sp. 182SMLY producing quinones exhibited strong antibacterial activity against MRSA [106]. Medermycin-type naphthoquinone-streptoxepinmycin A to D derived from the marine *Streptomyces* sp. XMA39 displayed antibacterial and antifungal activities against *S. aureus*, *E. coli*, and *C. albicans* [107]. As a result of these findings, it appears that marine bacteria create relatively more significant pigments with antimicrobial properties [108]. *Dunaliella* spp. is well-known for creating bioactive pigments from their methanol and chloroform extracts against pathogens, such as *B. subtilis* and *E. coli*, with inhibition zones measuring 20, 19, 18, and 22 mm, respectively. Through GC-MS and HPLC-DAD analyses, the chloroform extract of *Dunaliella* sp. 2 containing active pigments, such as luetin, carotene, and Zeaxanthin, was proven to have the aforementioned activity [109]. *Dunaliella* sp., which produces orange-red pigments, showed antibacterial and antiviral properties [110].

### 6.2. Biosurfactants

The partially purified biosurfactants containing compound 1, 2-Ethanediamine N, N, N′, N′ -tetra, 8-Methyl-6-nonenamide, (Z)-9-octadecenamide, and fatty acid derivatives retrieved from *Halomonas* sp. BS4 showed activity against human pathogens, including *S. aureus* (15 mm), *K. pneumoniae* (15 mm), and *S. typhi* (17 mm), and growth inhibition on the fungus *A. niger* [31]. The same team discovered halophilic *Bacilllus* sp. BS3 in Kaniyakumari, India, which produced a lipopeptide biosurfactant comprising compounds such as 13-Docosenamide., (z); Mannosamine,9-; and N,N,N′,N′-Tetramethyl and showed antiviral activity against the White spot syndrome virus (WSSV) by suppressing viral replication at their higher concentrations of 50%, 75% and 100%, respectively. The aforementioned purified biosurfactants were found to have antibacterial activity against *E. coli* and *S. aureus* at 20 µL concentrations, with inhibition zone diameters of 16.0 and 14.06 mm, respectively. Alvionita and Hertadi (2019) [111] conducted an intriguing investigation employing *Halomonaselongata* BK-AG18 to bioconvert glycerol into a biosurfactant in a nutritional medium with glycerol as the sole carbon source at an optimal pH 6. The growth inhibition efficacy of a purified biosurfactant was observed against *S. aureus* at 1000 mg/L by reducing its optical density (OD_600_). The biosurfactants produced from halophilic bacteria, such as *Halomonaselongata*, *Halobacilluskarajiensis*, and *Alkalibacillusalmallahensis*, proved its antimicrobial activity at a 100 µg/mL concentration by reducing the OD value on *S. aureus* (97.75%), *Enterococcus faecalis* (97.6%), and *B. subtilis* (97%) [9]. Antimicrobial glycolipid biosurfactants were recovered from the halophilic bacterium *Buttiauxella* sp., isolated from soils of the Qeshm Island mangrove forest, southern Iran. The antimicrobial activity of the produced biosurfactants was confirmed against the pathogens *B. cereus* (250 µg/mL), *E. coli* (200 µg/mL), *S. enterica* (250 µg/mL), *B. subtilis* (300 µg/mL), *A. niger* (100 µg/mL), and *C. albicans* (150 µg/mL) [49]. *Pseudomonas* sp., isolated from a polluted saltpan, Puthalam district, Kanyakumari, developed biosurfactants with high antibacterial activity to Gram-negative strains *E. coli* (15 mm), *K. pneumoniae* (13 mm), and *V. cholerae* (10 mm) [112]. An interesting report says the anti-biofilm activity of a biosurfactant produced from *Halomonas* sp. isolated from the sediments of the Bay of Bengal showed 99.8% growth inhibition on *S. typhi* and 99.5% on *V. cholerae* at 125 g/mL Con [113]. A new biosurfactant named leu/ile-7 C15 surfactin [M + Na]+ derived from the moderate halophilic bacterium *B. tequilensis* ZSB10 isolated from Crystal salt pond, Las Ventas, showed antifungal action by growth inhibition of *Helminthosporium* sp. at 79.3% and also an IC50 at 1.37 mg/disc [114]. The biosurfactant produced from *Halobacterium salinarum* showed antimicrobial activity against *Bacillus* spp., *Streptococcus* spp., *E. coli*, *Pseudomonas* spp., *S. aureus*, *C. albicans*, and *A. niger* [115].

### 6.3. Exopolysaccharides

Marine bacteria produceexopolysaccharides (EPS) with various sugar and non-sugar compounds such as arabinose, xylose, glucose, acetic acid, and succinic acid from *Bacillus*, *Alteromonas*, *Pseudoalteromonas*, and *Vibrio* species that possess several pharmacological properties, including antimicrobial responses [116]. Several marine bacterial supernatants were shown to exhibit anti-biofilm activity by generating active chemicals ranging from furanones to multifunctional polysaccharides that were shown to be QS (Quorum sensing) inhibitors [117]. The marine *Bacillus altitudinis* MSH2014 isolated from mangrove sediments in Ras Mohamed, Red Sea Coast, Egypt, was able to produce mannuronic acid, glucose, and sulphate-containing heteropolysaccharide that gave an antimicrobial response against *B*. *subtilis* (17.8 mm), *S. aureus* (18.8 mm), *E. coli* (24.9 mm), *P. aeruginosa* (15.6 mm), and yeast, as well as fungi, including *S. cerevisiae* (17.6 mm), *C*. *albicans* (17.3 mm), *A. niger* (20 mm), and *F. oxysporum* (10.5 mm) at 200 µg/disc [118]. *Halomonassaccharevitans* AB32 were able to produce EPS at the optimum temperature of 25 °C and pH 9 using lactose and malt extract as their carbon and nitrogen sources with maximum EPS yields at 138 gL^−1^. The antimicrobial activity of the produced EPS was examined against the pathogenic bacteria *V. fluvialis* and the fungus *A. niger* by growth inhibition at the maximum absolute units of 14.1 and 25.1 [119]. Raffinose carbohydrate was significantly present in the HPLC analysis for the aforementioned EPS with a significant peak at a retention time of 3.910. Halophilic species such as *Bacillus*, *Halomonas*, *Psychrobacter*, and *Alcaligenes* produced eight EPS compounds with antimicrobial efficacies, and E15 strains were reported to be more active against *B. cereus*, *S. aureus*, *S. saprophyticus*, *Enterobacter cloacae*, *Proteus mirabilis*, MRSA, *Enterococcus faecalis*, *Streptococcus pneumonia*, *Acinetobacter* sp, and *Campylobacter jejuni* with MICs ranging between 250 and 500 µg/mL [120]. E37 also exhibited a wide antimicrobial activity with 250, 62.5, 125, and 500 µg/mL MICs, respectively, against the same pathogens mentioned above. Generally, EPS produced from halophilic isolates displayed more antibacterial action from the genera *Halomonas*, *Chromohalobacter*, *Salinivibrio*, *Nesiotobacter*, *Brevibacterium*, *Virgibacillus*, and *Salinicoccus* against *E. coli, S. pasteuri*, *B. cereus, P. aeruginosa, M. luteus*, and *S. cerevisiae* [89]. According to the literature, a large number ofEPS were produced in saline areas, but only moderate antibacterial activity against microbial pathogens was identified.

## 7. Strategies behind Halophiles for Bimolecular Adaptation to Extreme Habitats

Microbial metabolite secretions at challenging habitats, such as saline/hypersaline ecosystems, could promote adaptations through specific pathways [121]. Moreover, hypersaline environments denoting salinities of more than ≈35‰, where seawater might even show an oversaturation of salts [122,123]. Halophilic bacteria and eukaryotes exploit the salt-out strategy that excretes salts from the cytoplasm, and they either synthesise or accrue the de novo attuned solutes, including glycine betaine, and some zwitterionic compounds in bacteria, such as glycerol, and certain polyols in eukaryotes [123]. Halophiles adopt common strategies to avoid an excessive loss of water due to NaCl saturation. These include cellular adaptations, high salt-instrategy or low salt/solute-instrategy (Figure 3). The first one produces osmoprotectants that increase osmotic-cytoplasm activity to adjust to the external environment or reach the equilibrium state by increasing high salt concentrations so that their cytoplasm matches with high environmental salt concentrations. The high salt-in strategy performs the protection of halophiles through the accumulation of inorganic solutes intracellularly to balance the salt concentration of the external environmentthrough the uniport and symport system in the presence and absence of light. In the third strategy, osmolytes from the external environment protect the cell protein from denaturation [124]. The adaptations of halophilic biomolecules are documented through various mechanisms. Especially in fungi, the glycerol signalling pathway with high osmolarity to increase the salt level was screened between the fungus *W. ichthyophaga* and *H. werneckii* as halotolerant/halophilic fungi [125]. Even some halophilic protists express high gene proportions in duplicated genes at high salt concentrations that were expressing different levels in *H. seosinensis*, which has its acquisition from bacteria that could evidence the evolutionary process that might facilitate high salt adaptation [126]. Halophilic metabolite production could depend on salinity, as evidenced for *Bacillus* VITPS3, which produced 3.18-fold more metabolites in the presence of 10% (*w*/*v*) NaCl from various tested concentrations [127]. Moreover, in media enrichment apart from salinity, the source of carbon has its potential towards antimicrobial production in the culture media [14]. The role of salinity in halophilic and halotolerant microbes might vary since halotolerants can grow in the presence but also in the absence of higher salt concentrations, which was recently shown for *Exiguobacterium* sp. SH31, which can grow in up to 50 g/L of NaCl [128]. In order to produce potential metabolites from complex halophiles due to various salinity gradients, recent strategies such asthe mixed culture approach were developed by Conde-Martinez et al. (2017) [53]. It is used to isolate potential strains from different ecosystems, including brine and sediment samples via inoculation into different media, and to finally obtain an organic extract to screen for antimicrobial activity. Metagenomic applications such as the sequencing of 16s rRNA illumine amplicon were applied in Karak mine salterns, Pakistan. Here, 66% of the bacterial consortia occurred in brine, and 72% from salt regions were dominated by Bacteroidetes and Proteobacteria with a high abundance of Archaea [129]. Hence, metagenomics demonstrated an efficient approach to address the bioactive microbial species at different saline habitats.

## 8. Applications and Future Perspectives of Halophiles as Pharmaceuticals

Halophilic microbial products are predicted to have significant uses in the pharmaceutical sector and healthcare [130]. Proteolytic enzymes are used to produce pharmaceutical products [131]. According to bioactive compounds, diverse halophilic bacteria are employed to produce bioactive compounds, which are significant and understudied sources of bioactivities, such as antiviral, antibacterial, and anti-tumour agents [38]. Figure 4 depicts the structure of different antimicrobial compounds produced by halophilic microbes generated by the ChemDraw (Version 20.1.1) drawing tool. Marine cyanobacteria have gained a lot of attention as a powerful group in the creation of pharmaceuticals such ascryptophycin and curacin, which are currently in clinical trials [132]. Peptide molecules from marine diatoms also have been explored with respect to their antioxidant and anticancer properties [133,134]. Biosurfactants from halophiles are receiving more attention for antioxidants, antiviral antibacterial, antifungal, anticancer, antiviral, anti-adhesive, immunomodulator, stimulating dermal fibroblasts, gene therapy, and vaccines [135]. Halophilic bacteria must reach a tipping point in the future by manufacturing various novel drugs, antioxidants, sunscreens, compatible solutes, and hydrolytic enzymes from unexplored regions. Recent advances in the incorporation of halogenated compounds into peptoids (oligomers of N-substituted glycines) improve antimicrobial efficacy against multi-drug-resistant pathogens, with brominated analogues showing 32-fold increased activity against MRSA and 16-64-fold increased activity against *P. aeruginosa* and *E. coli* [136]. In the future, halogenated drugs may have increased action against drug-resistant bacteria [136,137]. OMIC technologies present new potential for the discovery of exclusive properties and/or novel biomolecules derived from halophiles in the future [138,139] as a result of recent findings of halophilic bacteria, even from terrestrial environments [140].

Further research is needed to report on how halophilic microorganisms evolved during the early phases of evolution of life on earth, as well as how they diversified and spread around the world. Their biotechnological potency for generating compatible solutes, biopolymers, and other molecules is of industrial interest. To fully realise their clinical potential, additional research must focus on their physical organisation and modes of action, allowing physicians to forecast which molecule could produce the desired medicinal effect.

## 9. Conclusions

Researchers focusing on halophilic ecosystems in their search for novel biomolecules are mostly motivatedby the threat of drug-resistant human pathogens. This review highlights that no Haloarchaeon has been found to show antibacterial action. More new compound extraction from more halophilic microbial genera is needed to combat human pathogenic drug-resistant microbes. Halophilic representatives of *Bacillus* and the dominating actinomycete biomolecules have already been demonstrated to be effective against human drug-resistant infections. There is no benign report yet for the enzymes from halophilic microbes against human pathogens. However, clinical trials should focus more on antimicrobials produced from halophiles because knowledge on the mode of action of halo-antimicrobials against drug-resistant organisms is lacking. Overall, this short review summarises the risk of clinical drug-resistant strains and signifies its control using halo-derived compounds as a more promising strategy.

## Figures and Tables

**Figure 1 microorganisms-10-00417-f001:**
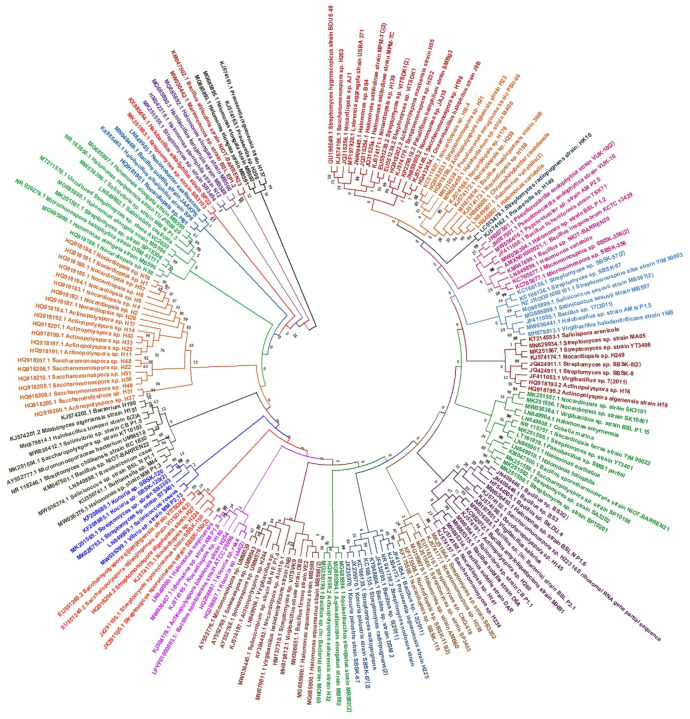
Phylogenetic representation of halophilic bacterial genera producing antimicrobial metabolites, as computed from recent literature (after 2010).

**Figure 2 microorganisms-10-00417-f002:**
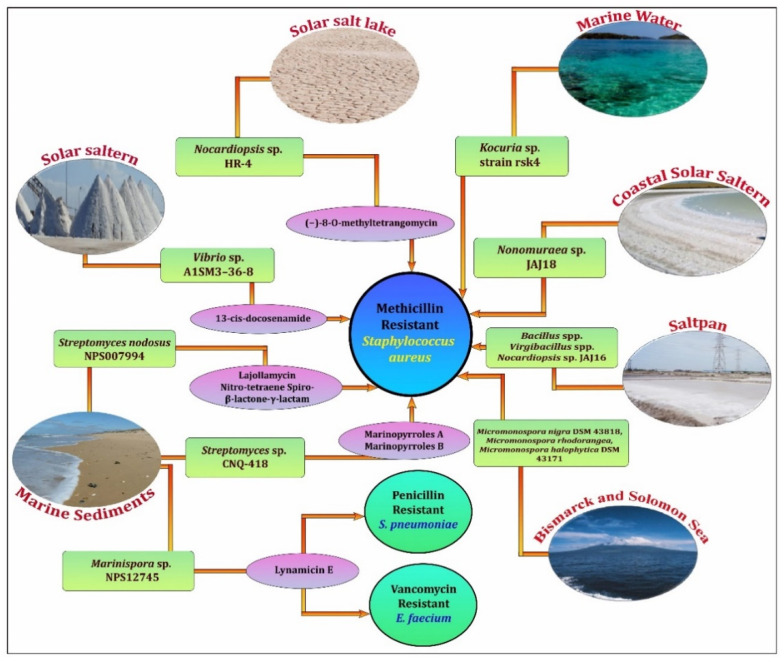
Representation of diverse halophilic ecological metabolites reported against drug-resistant pathogens.

**Figure 3 microorganisms-10-00417-f003:**
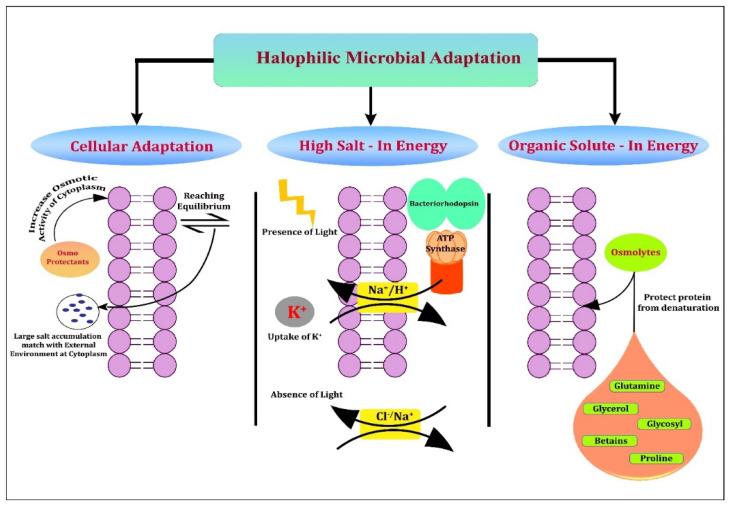
Common strategies involved in halophilic microbial adaptations consisting of cellular adaptation, high salt-in energy, and organic solute-in energy.

**Figure 4 microorganisms-10-00417-f004:**
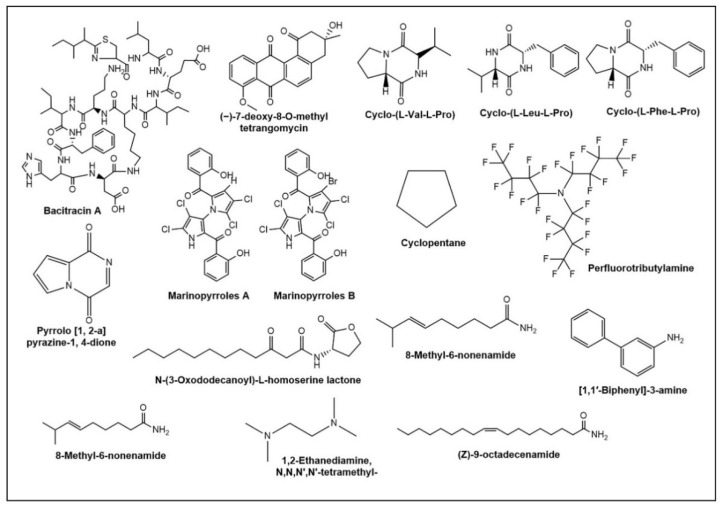
The structure of different antimicrobial compounds produced by halophilic microbes.

**Table 1 microorganisms-10-00417-t001:** Produced antimicrobials from halophilic microbes against different clinical pathogens.

S.No	Organism	Isolation Source	Compound	Activity	Reference
1.	*Bacillus* sp.	Condenser water, solar salt works in Thamaraikulam, Kanyakumari district, Tamil Nadu, India	13-Docosenamide, 9-Octadecenamide, Cylohex-1,4,5-triol-3-one-1-carbo	Antibacterial and Antifungal	[31]
2.	*Halomonassalifodinae*	Solar salt condenser, Thamaraikulam solar astern, Kanyakumari district, Tamil Nadu, India	Perfluorotributylamine, Pyridine, 4-(phenylmethyl), Nonadecane	Antibacterial	[32]
3.	*Pseudonocardiaendophytica*	Sediments of mangrove Nizampatnam, Bay of Bengal, Andhra Pradesh, India	3-((1H-indol-6-yl) methyl) hexahydropyrrolo [1,2-a] pyrazine-1,4-dione	Antibacterial	[43]
4.	*Piscibacillus* sp.	Sambhar Lake in India	Crude extract	Antibacterial and anticancer	[12]
5.	*Nocardiopsis* sp.	Saline soil of Kovalam solar salterns India	Pyrrolo (1,2-A (pyrazine-1,4-dione, hexahydro-3-(2-methylpropyl)-)	Antibacterial	[13]
6.	*Nocardioides* sp.	Antarctic Casey Station, Wilkes Land,	Glycolipids and/or lipopeptides	Enzymatic and antimicrobial activities	[14]
7.	*Aspergillus flocculosus*	Putian saltern of Fujian, China	6-(1H-pyrrol-2-yl) hexa-1,3,5-trienyl-4-methoxy-2H-pyran-2-one	Antibacterial	[44]
8.	*Bacillus subtilis, Bacillus licheniformis*	Halophilic MaharluSalt Lake—Iran	glycoprotein	Antifungal, Antibacterial	[22]
9.	*Virgibacillusmarismortu*, *Terribacillushalophilus*	Halophilic Tunisian Sebkha	Glucanase, thermotolerant chitinases	Antimicrobial activity, Antifungal enzymes	[45]
10.	*Nocardiopsis terrae*	Saline soil, Qaidam Basin, north-west China	Quinoloid alkaloid 4-oxo-1,4-dihydroquinoline-3-carboxamide, Indole-3-carboxylic acid	Antibacterial and anticancer	[46]
11.	*Aspergillus flavus*, *Aspergillus gracilis*	Solar saltern, Phetchaburi, Thailand	Crude extracellular compounds	Antibacterial and antioxidant	[40]
12.	*Halomonas* sp.	Halophilic bacteria Yuncheng Salt Lake, China	Amylase, protease, lipase, cellulase, pectinase and DNAase	Antimicrobial activity, hydrolytic activities.	[23]
13.	*Streptomonosporaalba*	Soil sample, Xinjiang Province, China	Streptomonomicin	Antibacterial	[47]
14.	*Salinisporaarenicola*	Great Barrier Reef (GBR) sponges, Queensland, Australia	Rifamycin B, S and W	Antifungal	[48]
15.	*Nocardiopsis* *lucentensis*	Salt marsh soil, Alicante, Spain	Nocarbenzoxazole G	Antibacterial and anticancer	[49]
16.	*Buttiauxella* sp.	Halophilic, marine bacteria mangrove forest, Qeshm Island, south of Iran	Glycolipid biosurfactant	Antimicrobial activity	[50]
17.	*Actinomyces* sp.	Halophilic AranBidgolandMaharlu Lakes in center and south of Iran	Chloroacetate, ethylcholoroacetate and 4-chloro-3hydroxybutyronitrite groups	Antimicrobial activities	[51]
18.	*Paludifilumhalophilum*	Sfax solar saltern, Tunisia	Gramicidin S, Cyclo(l -4-OH-Pro- l -Leu), Cyclo(l -Leu- l -Pro)	Antibacterial	[52]
19.	*Vibrio* sp.	Brine and sediments from Manaure solar saltern. La Guajira, Colombia	13-cis-docosenamide	Antibacterial	[53]
20.	*Nocardiopsis* sp.	Salt lake soil, Algerian Sahara. Algeria	Compound 1:(−)-8-O-methyltetrangomycin	Anticancer	[54]
21.	*A. protuberus*	Arctic sub-sea sediments from the Barents Sea	Bisvertinolone	Antifungal	[15]
22.	*Bacillus* sp.	Halophilic	carotenoids, polyhydroxy alkanoates, ectoine, bioplastics and enzyme	Antibacterial Activity	[55]
23.	*Halomonas elongate*, *Halobacilluskarajiensis, Alkalibacillus* *almallahensis*	Halophilic extreme saline soil samples of Khewra Salt Mines, Pakistan	Peptide furanomycin, biosurfactants	Radical scavenging activity, antioxidant potential, antimicrobial activity	[9]
24.	*Halomonaselongata*	Halophilic	Ectoine	Antimicrobial activity	[56]
25.	*Coccomyxaonubensis*	Tinto river, Spain	Palmitic acid, oleic acid, linoleic acid	Antibacterial and Antifungal	[57]

## Data Availability

Not applicable.

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
