# Peer review of "Recent Antimicrobial Responses of Halophilic Microbes in Clinical Pathogens"

_microorganisms, 2022, doi:10.3390/microorganisms10020417_

Round 1
Reviewer 1 Report
Dear Authors,
Thank you for trying to follow my recommendations. There is an improvement of the manuscript.
I would have the following recommendations:
- please revise (thoroughly) the English
- please rephrase the title. In its present form it is very confusing (at least in my opinion) and hard to understand.
- line 160 - the sentence is very misleading, since it is formulated as the drugs are given as examples for Gram negative bacteria
- the whole manuscript should be written more concise
Author Response
Q1; Thank you for trying to follow my recommendations. There is an improvement of the manuscript. I would have the following recommendations: please revise (thoroughly) the English
Response: Thank you very much for the appreciation. The English was checked thoroughly with the entire manuscript as per the reviewer suggestions and modified.
Q2; Please rephrase the title. In its present form it is very confusing (at least in my opinion) and hard to understand.
Response: Thanks for the suggestion and the title was changed to “Recent antimicrobial responses of halophilic microbes in clinical pathogens”
Q3; Line 160 - the sentence is very misleading, since it is formulated as the drugs are given as examples for Gram negative bacteria
Response:The line 160 was only given for the example of drug resistance in both gram negative and positive bacterial strains and now it is shortened and changed in the revised manuscript in the red font by following the reviewer comments.
Q4; - The whole manuscript should be written more concise
Response: The whole manuscript is now written more concisely following the suggestions of the reviewer.
Reviewer 2 Report
In this manuscript, the authors review the state of art regarding the discovery of new bioactive molecules produced by halophilic microorganisms. Although I think it is a well-done manuscript, some critical issues need to be addressed.
First of all, I suggest a thorough revision of the language; it is challenging to understand what the authors mean in some sections. I recommend the use of synonyms to avoid repeating the same words many times in the text.
Secondly, I suggest drastically increasing the quality and resolution of the figures; for example, I find figure 1 practically unreadable.
As this is a review, I suggest making a small conclusion at the end of each paragraph.
It would be helpful to elaborate on the antimicrobial action of pigments, biosurfactants, etc., in paragraph 6.
Author Response
Q1; In this manuscript, the authors review the state of art regarding the discovery of new bioactive molecules produced by halophilic microorganisms. Although I think it is a well-done manuscript, some critical issues need to be addressed. First of all, I suggest a thorough revision of the language; it is challenging to understand what the authors mean in some sections.
Response: Thanks for the comments and the English is now checked thoroughly in the entire manuscript and modified in the revised version.
Q2; I recommend the use of synonyms to avoid repeating the same words many times in the text.
Response: Appropriate synonyms were added to avoid repetition of words in the text.
Q3; Secondly, I suggest drastically increasing the quality and resolution of the figures; for example, I find figure 1 practically unreadable.
Response: Thanks for the comments. The figure resolution is improved and now it’s in readable form by following the suggestion of reviewers.
Q4; As this is a review, I suggest making a small conclusion at the end of each paragraph.
Response: Small conclusions were added appropriately in the revised version in the red font.
Q4; It would be helpful to elaborate on the antimicrobial action of pigments, biosurfactants, etc., in paragraph 6.
Response: Thanks for the comments. To section 6 of the revised version some more elaborated antimicrobial responses of halo-derived bio-surfactants, pigments and EPS was added by following the comments of the reviewer was given in the red font.
Round 2
Reviewer 1 Report
Dear authors,
The manuscript was improved. The title is shorter – but not clear, yet.
If you really want to use this one, please explain in a separate sentence (answer) the reason for: „antimicrobial responses of halophilic microbes in clinical pathogens”.
There are authors that use Enterobacteriaceae and not Enterobacterales.
I selected some long phrases. Please evaluate for rephrasing. Even better, another English revision by a „mother tongue” person would be good.
„On the other hand, according to a recent surprising report, Fluconazole resistance in Candida albicans was induced by the quorum sensing (QS) of P. aeruginosa by producing QS component N-(3-Oxododecanoyl)-L-homoserine lactone, inducing
resistance in Candida spp. to fluconazole via the reverse pathway of ergosterol biosynthesis.”
„Halophilic Antarctic soils containing Nocardioides sp., the cell supernatants of the bacterium containing glycolipids and/or lipopeptides provided antimicrobial efficacy, whereas salt medium supplemented with various carbon sources provided enzymatic activity as well as a broad spectrum of antimicrobial activity against S. aureus and X. oryzae [14].”
„The novel halophilic strains AH35 and AH10 isolated from the saline soils of the Algerian Sahara showed antibacterial activity specifically against gram negative rather than gram positive strains including K. pneumoniae, Pseudomonas syringae, and Agrobacterium tumefaciens with varied 13-45 mm diameter of inhibition zones also AH35 demonstrated the sensitivity of Salmonella enterica (13mm), these putative strains may belong to novel species such as Saccharomonospora paurometabolica, Saccharomonospora halophila, and Actinopolyspora iraqiensis, according to phylogenetic clades [95].”
Author Response
Reviewer report 1
Q1; The manuscript was improved. The title is shorter – but not clear, yet.
If you really want to use this one, please explain in a separate sentence (answer) the reason for: „antimicrobial responses of halophilic microbes in clinical pathogens”.
Response; The title simply says that ’the antimicrobial property/responses of halophilic microbes against human pathogens are harmful”. With antimicrobial responses we mean the antimicrobial activity and with recent we mean that the information were used in the review mostly covered the period from 2010-2021. Considering the above aspects we changed the title to “Recent antimicrobial responses of halophilic microbes in clinical pathogens’’.
Q2; There are authors that use Enterobacteriaceae and not Enterobacterales.
Response; Due to changes of bacterial systematics, the Enterobacterales became a new scientific order. Enterobacteriaceae are currently included in the Enterobacterales order along with species belonging to the Erwinaceae, Yersiniaceae etc. But we won’t signify that species usage in this review. Most commonly E. coli, Klebsiella and Salmonella strains were noted in this review that belong to the large family of the Enterobacteriaceae.
Q3; I selected some long phrases. Please evaluate for rephrasing. Even better, another English revision by a „mother tongue” person would be good.
„On the other hand, according to a recent surprising report, Fluconazole resistance in Candida albicans was induced by the quorum sensing (QS) of P. aeruginosa by producing QS component N-(3-Oxododecanoyl)-L-homoserine lactone, inducing
resistance in Candida spp. to fluconazole via the reverse pathway of ergosterol biosynthesis.”
Response; This sentence has been ephrased in the revised manuscript.
„Halophilic Antarctic soils containing Nocardioides sp., the cell supernatants of the bacterium containing glycolipids and/or lipopeptides provided antimicrobial efficacy, whereas salt medium supplemented with various carbon sources provided enzymatic activity as well as a broad spectrum of antimicrobial activity against S. aureus and X. oryzae [14].”
Response; Sentence has been rephrased in the revised manuscript as made visible by review mode.
The novel halophilic strains AH35 and AH10 isolated from the saline soils of the Algerian Sahara showed antibacterial activity specifically against gram negative rather than gram positive strains including K. pneumoniae, Pseudomonas syringae, and Agrobacterium tumefaciens with varied 13-45 mm diameter of inhibition zones also AH35 demonstrated the sensitivity of Salmonella enterica (13 mm), these putative strains may belong to a novel species such as Saccharomonospora paurometabolica, Saccharomonospora halophila, and Actinopolyspora iraqiensis, according to the phylogenetic clades [95].”
Response; Sentence has been rephrased in the revised manuscript as made visible by review mode.
Reviewer 2 Report
The manuscript, after corrections, has improved a lot.
A small problem remains in figure 2 where the line numbers are within the figure.
Author Response
Reviewer report 2
Q1; The manuscript, after corrections, has improved a lot.
A small problem remains in figure 2 where the line numbers are within the figure.
Response; Thank you very much for the appreciation. Actually in the word file that Figure 2 image was fine and clear. In the PDF version (building PDF while uploading) only the line numbers are merging with the image at the left.
This manuscript is a resubmission of an earlier submission. The following is a list of the peer review reports and author responses from that submission.
Round 1
Reviewer 1 Report
Dear authors,
The subject and the title is interesting and for sure a lot of work was involved.
Even though my assumption is correct, some improvement is needed. Please do follow the title. In some parts of the manuscript the link between the title and the text is not obvious. The best link started with figure 1, table 1 and item 4. Please think to reduce the introduction to clear paragraph, in line with the chosen subject. Please re-organize item 1 (introduction) as well as items 2-3. For some information you could use a synthetic table.
Please clarify if figure 2 is original and the software used to prepare it.
As discussed, table 1 is a quite good selection of information. Please clarify your choice for 50, 55, 69, 72, 75-80 and 82.
The reference list is a long one. Just like some examples (from the pages 1 and 2 of References) it is not obvious your choice for items: 1, 3, 4, 6, 13, 15, 23 (being old or not so relevant). To be more explicit: why to choose fact sheets from WHO and CDC (there are technical and scientific better documents; even on their sites).
Please do recheck the phrases, if possible with a English native colleague (like some examples: Microbial pathogens causing severe infections and being drug resistant are getting more virulent at the same time. p1, Usage of large antibiotics on human therapy as well as to animals such as for farm
animals including fishes results in pathogenic microbial selection that caused resistance to multiple drugs. p1
Please do recheck the spelling (e.g. Vibrio cholera - p2)
Reviewer 2 Report
The main goal of this paper is to perform a review about the antimicrobial compounds retrieved from microbial sources of various saltern environments and to evaluate their potential against clinically drug resistant bacteria.
Major comments: All manuscript is very difficult to read and understand. The English should be carefully revised. I recommend a reduction in the length of the sentences and the inclusion of a reference in all sentences. Nevertheless, this is an interesting revision about halophilic microbes. New research focusing on antimicrobial resistance and new antimicrobial drugs for fighting this problem are always important topics.
Title: What do you mean by advanced antimicrobials???
Introduction section: This section is confused. I recommend an English revision, but also some reformulations:
- Lines 42-43 “ Clinical sectors have been confronted with risky challenges, and its emergence seeks improved antibiotics, particularly against human pathogens which cause serious threads 43 [1,2].” When you say risky challenges several issues may be included, so it is not accurate to say that the emergence of risky challenges needs improved antibiotics. I don’t know if these risky challenges are economical, for example. You never referred that these risks are health challenges.
- Then Lines 44-46, it is not the use of antibiotics for itself that promotes antimicrobial resistance dissemination, It’s the misuse and overuse of them. Please correct that.
- Line 48 what do you mean by “antimicrobial bacteria or fungi”? I think you wanted to say antimicrobial resistant bacteria or fungi.
- Line 49 ABR????
- Line 51 what do you mean by “ advanced drug treatment”?? it is not clear.
- Line 52-53 “However, inclusive killing and bring to an end of the drug resistance phenomenon is extremely difficult in the human environment [5].” I do not understand the meaning of inclusive killing. The drug resistance phenomenom is difficult solve in the human, animal and environmental sectors and not only in Human sector. Reformulate the sentence.
- Reference 16 in line 77 is not accurate. Reference 16 does not mention MRSA, as is described in the manuscript. Please verify this point. You may probably see this reference: Lee DS, Eom SH, Jeong SY (2013) Anti-methicillin-resistant Staphylococcus aureus (MRSA) substance from the marine bacterium Pseudomonas UJ-6. Environ Toxicol Pharmacol 35:171–177
- Line 79-80 what do you mean by “ pathogenic drug resistance 79 reduction of halophilic strains”
Section 2 Halophiles:
Lines 104-108 “Moreover, solar saltern Halomonas salifodinae secretes Perfluorotributylamine, Cyclopentane, 1-butyl-2-ethyl and 1,1′-Biphenyl]-3-amine antimicrobial metabolites provides more than a 10 mm zone of inhibition against aquatic pathogens such as Vibrio parahaemolyticus, Vibrio harveyi, Aeromonas hydrophila and Pseudomonas aeruginosa isolated from fish and shrimp.” This sentence needs a reference, and it needs to be corrected. Which of the metabolites provide inhibition of the mentioned bacteria? What is the meaning of 10 mm zone of inhibition? It is an inhibition zone diameter?? Reformulate the sentence.
Line 110 when you say the same compound you refer to which of the 3 metabolites? Or all them? It is not clear.
Line 115 EPS???
Line 119 “numerous” correct to Numerous
Line 124 to 126 the sentence is confused I do not understand its meaning.
Line 130 remove “ have” and add it in line 131 before “been”
Line 134 include reference
Section 3:
Lines 144-145 Confused sentence. Reformulate please.
Line 165 include reference
Line 171 “….could trait depends…” confused!!! Reformulate please.
Section 4:
Lines 195 to 203 are confuse. Reformulate please and reduce the sentence.
Line 206 You talk about non-susceptibility to penicillin but before that you always use the word resistant, for the other strains. Why is that?
Line 211 “Moreover, halophilic biomolecules against drug resistance…” I believe you wanted to say: Moreover, halophilic biomolecules against drug resistant bacteria
Line 212 include a reference
Line 217-219 “The halophilic Bacillus provides considerable biomolecules but most of them attribute anticancer agents than antimicrobials [36].” Confused sentence. Maybe you wanted to say …. most of them are considered anticancer agents rather than antimicrobials
Line 224 S. aureus
Line 225 to 232 – I do not understand the meaning of the sentences.
Line 227 – You need to include the meaning of MIC and ug/mL should be corrected to µg/mL
Line 249 – there is a repetition of H262 ( please remove it)
Line 251-252: What do you mean by 17mm growth inhibition? It is an inhibition zone diameter or what method have been used?
Line 254 to 257 confused.
Section 5 :
Line 271 include reference
Line 273 OD???
Line 281 Do you really wanted to say impotency?
Line 296 and 299 include reference
Section 6:
Line326: ug/mL should be corrected to µg/mL
Section 7:
Line 338- “…of Mucor spp.” correct to against Mucor spp.
Line 341 – correct includes to including
Line 343-347 confused. Please reformulate.
Section 8:
Lines 365, 372 and 375 include references
Line 388 – Correct Bacillus to Bacillus